# Inhibitory Properties of Aldehydes and Related Compounds against *Phytophthora infestans*—Identification of a New Lead

**DOI:** 10.3390/pathogens9070542

**Published:** 2020-07-07

**Authors:** John J. Mackrill, Roberta A. Kehoe, Limian Zheng, Mary L. McKee, Elaine C. O’Sullivan, Barbara M. Doyle Prestwich, Florence O. McCarthy

**Affiliations:** 1Department of Physiology, School of Medicine, University College Cork, T12 K8AF Cork, Ireland; limian.zheng@ucc.ie; 2School of Chemistry and Analytical and Biological Chemistry Research Facility, University College Cork, Western Road, T12 K8AF Cork, Ireland; 118225550@umail.ucc.ie (R.A.K.); m.mckee@umail.ucc.ie (M.L.M.); elosullivan@hotmail.com (E.C.O.); 3School of Biological, Earth and Environmental Sciences, University College Cork, T12 K8AF Cork, Ireland; b.doyle@ucc.ie

**Keywords:** *Phytophthora infestans*, cinnamaldehyde, thiosemicarbazone, ellipticine, fungicide

## Abstract

The pathogen *Phytophthora infestans* is responsible for catastrophic crop damage on a global scale which totals billions of euros annually. The discovery of new inhibitors of this organism is of paramount agricultural importance and of critical relevance to food security. Current strategies for crop treatment are inadequate with the emergence of resistant strains and problematic toxicity. Natural products such as cinnamaldehyde have been reported to have fungicidal properties and are the seed for many new discovery research programmes. We report a probe of the cinnamaldehyde framework to investigate the aldehyde subunit and its role in a subset of aromatic aldehydes in order to identify new lead compounds to act against *P. infestans*. An ellipticine derivative which incorporates an aldehyde (9-formyl-6-methyl ellipticine, **34**) has been identified with exceptional activity versus *P. infestans* with limited toxicity and potential for use as a fungicide.

## 1. Introduction

One of the most significant threats to global food production and agriculture is the pathogenic oomycete *Phytophthora infestans* (*P. infestans*) [1]. It was first described by Berkeley in 1846 and subsequently named *Phytophthora infestans* by deBary in 1876, which aptly translates as “plant destroyer” [2,3]. Most infamously known for the Irish potato famine in 1845–1849, the pathogen caused incessant, devastating outbreaks of late potato blight resulting in the deaths of one million people and a further one million emigrating [4]. The current global impact of *P. infestans* is in excess of $6.2 billion per annum due to crop loss and the use of fungicides [5]. The necessity of fungicides to combat the spread of *P. infestans* is evident and with resistant strains continually developing, designing novel fungicides to target this pathogen is of high importance.

The Irish Famine of the 1840s spurred the development of the first-ever pesticide to be adopted worldwide. Developed by French scientists, Pierre Millardet and Ulysse Gayon, the Bordeaux mixture was used to combat the *P. infestans* blight. This fungicide is a mixture of copper sulphate and lime, which forms the active compound copper(II) hydroxide, stabilised by calcium sulphate. The emergence of this fungicidal treatment was imperative to the conservation of crops and averting hunger, therefore saving lives [6]. The use of the Bordeaux mixture diminished through the emergence of less toxic and more targeted fungicides [7]. 

The Bordeaux mixture was broadly used from the 1890s until it was superseded by the dithiocarbamate class of fungicides in the 1960s. The broad-spectrum fungicides were used due to their high biological and chemical activity, with the benefit of low production cost [8]. Mancozeb (**1**) and Propineb (**3**) were two of the most widely used dithiocarbamate fungicides [9,10]. Both of these organosulfur fungicides have been found to break down into toxic metabolites as degradation begins during storage to form the respective toxic metabolites (Figure 1) [11,12]. Ethylene thiourea **2** (R = H), one of the main metabolites of Mancozeb, has been shown to have many toxic effects in rats such as teratogenicity, developmental toxicity and promoting thyroid neoplasms [11]. Propylene thiourea **4** (R = CH_3_) from Proprineb has also been shown to affect the thyroid and the nervous system [8]. Propineb (**2**) has been banned by the European Commission and Mancozeb (**1**) is now under scrutiny for its negative effects. 

Following dithiocarbamates, phenylamide fungicides were developed in the 1970s. A significant advance in the control of *P. infestans* is attributed to the phenylamides. Metalaxyl (**5**) (Figure 2) is one of the most popular phenylamide fungicides, however, seven years after the product was commercialised, the first resistant strains of *P. infestans* were identified in Israel [13]. By 1980, resistant isolates of *P. infestans* were discovered outside of Israel, in Ireland, Switzerland and The Netherlands, demonstrating a need for further fungicidal development. To combat the growing resistance of *P. infestans* for phenylamides, a mixture was formulated with other fungicides (e.g., Mancozeb, **1**) which resulted in the reduction of resistant strains emerging [14].

Approved by the EU in 2017, oxathiapiprolin (**6**) has been used in more recent years as a targeted antioomycete fungicide (Figure 2). It was designed and synthesised by Pasteris et al. in 2016 [15] basing the structure around a piperidine-thiazole-carbonyl core. After numerous derivatives were synthesised and tested, the chemical structure of oxathiapiprolin (**6**) exhibited the greatest control over *P. infestans* with low toxicity in humans, birds and fish. The authors also investigated the mechanism of action of oxathiapiprolin and uncovered a novel target for oomycete inhibition. The compound binds to an oxysterol binding protein, although the role for this family is not very well understood. Oxathiapiprolin (**6**) has been approved in many countries around the world and requires a smaller quantity to be used on crops [16]. However, similarly to the phenylamides, oxathiapiprolin has already been found to be ineffective against emerging resistant strains of *Phytophthora capsici* [15,17]. The current focus has turned towards the use of natural products as fungicides to replace the previous synthetic alternatives. Although the first report of antimicrobial natural products was in 1676, development has been slow for these compounds [18]. Of prime interest for natural source fungicides is the essential oil, cinnamaldehyde.

Cinnamaldehyde (**7**, Figure 3) is a natural product extracted from the bark of the plant genus *Cinnamona*. This conjugated aromatic compound has shown to inhibit the growth of bacteria, filamentous moulds and yeast [19]. There are over 250 species of cinnamon plants and trees and are native to Sri Lanka and South India [20]. Cinnamaldehyde was first isolated from cinnamon oil in 1834 by Dumas and Peligot [21] and was first synthesised in 1854 by Chiozza [22]. The essential oil is generally recognised as safe by the United States Food and Drug Administration and has been used in foods and medicines as flavourings for many years [23]. It is also reported to possess many pharmacological activities such as antidiabetic and anti-inflammatory effects [24,25]. Extensive research has also been carried out on the antibacterial properties of cinnamaldehyde. The compound has demonstrated strong activity against many food-borne bacteria such as *Bacillus, Staphylococcus* and *Enterobacter spp.* and can therefore be used as a food preservative [26,27,28]. 

Studies by Quintanilla and Soylu [29,30] were some of the first to demonstrate the effect essential oils have on *P. infestans* and a follow up study by Ferhout et al. found that cinnamaldehyde (**7**) expressed a greater inhibitory effect on fungal strains than thyme oil [31]. Research by Raut determined cinnamaldehyde to be the most effective phenylpropanoid from plant origin against *Candida albicans* biofilm formation, being effective at completely inhibiting mature biofilms and identifying its potential [32]. Jantan et al. tested 14 essential oils against six dermatophytes, one filamentous fungus and five strains of yeasts, finding again that cinnamaldehyde possessed the strongest activity against all the different strains of fungi studied [33].

From experimental evidence, it appears cinnamaldehyde (**7**) inhibits cell wall biosynthesis, membrane function and some specific enzyme activities. However, specific cinnamaldehyde targets are still not well established. It is known that at lethal concentrations, the cell membrane is disrupted by cinnamaldehyde whereas at below lethal concentrations, it can inhibit ATPase and at low concentrations, it disrupts enzymes involved in cytokine action. It has been demonstrated in many studies that cinnamaldehyde (**7**) interacts with microbial cell membranes and that it is capable of altering the membrane lipid profile [34]. It is thought that the lipophilicity of cinnamaldehyde allows the molecule to partition through the lipid membrane of the cells and thus, disrupts bilayer structure. This renders the cell more permeable and prone to leakage of intracellular material causing cell death [26]. At lethal concentrations, cinnamaldehyde acts as a noncompetitive inhibitor of cell wall synthesising enzymes leading to the prevention of cell wall formation [35].

At sub-lethal concentrations, cinnamaldehyde inhibits transmembrane ATPase activity in *E. coli* and *Listeria monocytogenes* by entering into the cellular periplasm [36]. This enzyme is essential for maintaining transmembrane electrochemical proton gradient, necessary for the maintenance of cellular pH and uptake of nutrients [37]. At low concentration, cinnamaldehyde has been demonstrated to inhibit GDP dependent polymerisation, preventing cell division [38]. In *P.parasitica var. nicotianae,* Lu et al. found that cinnamaldehyde completely inhibited mycelial growth, sporangial formation, zoospore production and germination at concentrations ranging from about 10 μM to 50 μM [39].

In a study of resistance of *Capsicum annuum* (pepper plant) against root rot caused by *P.capsici*, Li et al. discovered that resistant plants displayed increased transcription of genes involved in the phenylpropanoid biosynthesis pathway, especially those involved in cinnamaldehyde production [40]. This suggests that cinnamaldehyde is a naturally occurring protective compound, whose production is enhanced in plant strains that are resistant to oomycete infections.

Transient Receptor Potential (TRP) ion channels play an important role in sensing damage to cells. Members of this channel superfamily, most closely related to the polycystic kidney disease (PKD) family, are encoded within the genomes of all oomycetes examined [41]. Compounds that are susceptible to nucleophilic attack from cystine can cause TRP channel activation, disrupting Ca^2+^ levels in the cell. Hu et al. demonstrated that the α, β-unsaturated bond in cinnamaldehyde is susceptible to nucleophilic attack from cystine, and therefore, modulates Ca^2+^ homeostasis in *Phytophthora capsici* (*P. capsici*) [42]. Intracellular free Ca^2+^ is a universal second messenger in eukaryotic cells, regulates cellular processes such as sporulation, germination and growth and is important in every life-cycle stage [41,43]. The covalently bound cinnamaldehyde to cystine was postulated to activate the TRP channel leading to an efflux of Ca^2+^ out of the cell and subsequently leads to cell death. To test their theory that the α, β-unsaturated bond in cinnamaldehyde is important, Hu monitored Ca^2+^ levels using a cinnamaldehyde derivative without the α, β-unsaturated bond, hydrocinnamaldehyde (**8**, Figure 3). Treatment of *P. capsici* with hydrocinnamaldehyde (**8**) demonstrated no inhibitory effects of zoospore growth indicating that the Michael addition through the α, β-unsaturated bond of cinnamaldehyde was crucial for stimulating Ca^2+^ efflux and growth inhibition of *P. capsici.*


Thus, although many studies exist on antimicrobial activity against several different groups of microorganisms, there is much to be learned about the mode of action of cinnamaldehyde **7** and the scope of its effects on *P. infestans*. As we set out to identify new agents against *P. infestans,* we initiated a broad study on the importance of the aldehyde aspect of cinnamaldehyde. 

Previous cinnamaldehyde analogues that have been subjected to *P. infestans* testing have primarily retained their aldehyde functionality and are substituted on the aromatic ring, examples of which are represented in Figure 4 (**9**–**13**) [44]. Separately, a recent study by Watamoto found a diverse set of compounds to demonstrate inhibitory effects on fungal strains, amongst which some of the structural features of our targets align [45]. Two hit compounds from this screen (**14** and **15**, Figure 4) have shown strong fungicidal effects and to inhibit the metabolic activity of *C. albicans*. Of these, Bay 11-7082 (**14**) possesses an α, β-unsaturated bond similar to cinnamaldehyde and may act through a similar mechanism. Ellipticine (**15**) has known antimicrobial and especially anticancer properties: testing at bacterial planktonic mode showed (**15**) had averaged a minimal inhibitory concentration (MIC) of 8.45 µM across all six *Candida* strains and hence is worthy of further investigation.

The aims of this study are to synthesise and evaluate cinnamaldehyde and aromatic aldehyde derivatives in assays of *P. infestans* growth and zoospore production. Cinnamaldehyde derivatives will be accessed by specific synthetic modification at positions R^1^, R^2^ and R^3^ represented in Figure 5. 4-Fluorocinnamaldehyde (R^1^ = F, **17**) was chosen to probe the effect on electron density withdrawal on the Michael acceptor. At the R^2^ position α-methylcinnamaldehyde (R^2^ = CH_3_, **18**) will test steric effects as the methyl group can influence attack on this reactive element. Finally, the rationale is provided by the structure of Mancozeb (**1**, Figure 5) for the generation of a hybrid structure **16** and the related thiosemicarbazone at the R^3^ position. Extension of the conjugated aldehyde will also be assessed to identify optimal chain length.

In order to probe the aldehyde functionality, a series of aromatic aldehydes and ellipticine aldehydes will also be assessed in the *P. infestans* assay to screen for potential lead compounds for future development (Figure 5). Given the experience of the group, of particular interest are the ellipticines which would offer a new framework for the inhibition of *P. infestans* growth and the potential for overcoming resistance.

## 2. Results

Synthesis of novel cinnamaldehyde and ellipticine derivatives was initially undertaken and in addition to some commercial aldehydes were assessed in a parallel approach, via the determination of inhibitory effects on the mycelial growth and zoosporogenesis of *Phytophthora infestans* and cytotoxicity against a human cell line. 

### 2.1. Effects of Mancozeb, Cinnamaldehyde and Hydrocinnamaldehyde on P. infestans Mycelial Growth

There is substantial published evidence indicating that Mancozeb (**1**) [9] and cinnamaldehyde (**7**) [39,42] inhibit the mycelial growth of *Phytophthora* and of other oomycetes. In contrast, at concentrations up to 2 mM, hydrocinnamaldehyde (**8**) is reported to have no detectable effect on the growth of mycelia from the zoospores of *P.capsici* [42]. We sought to test the potency of these molecules in inhibiting the mycelial growth in our experimental system (Figure 6). After five days of incubation, cinnamaldehyde displayed a minimal inhibitory concentration (MIC, the lowest concentration resulting in no visible growth) for *P. infestans* mycelial growth of approximately 1 mM and a half-maximal inhibitory concentration (IC_50_) of about 0.5 mM (see Appendix A). In contrast to the results of Hu et al. [42], we found that hydrocinnamaldehyde inhibited *Phytophthora* mycelial growth with an MIC of 2 mM and an IC_50_ of between 0.25 and 0.5 mM. However, it should be noted that the data reported by Hu et al. are from a different species (*P.capsici* versus *P. infestans*) and mycelia were generated from a different source (germinated from zoospores versus subcultured from growing mycelia). Mancozeb inhibited *P. infestans* growth most effectively, with an MIC of 100 μM and an IC_50_ of between 25 and 50 μM. These MIC and IC_50_ values for Mancozeb are similar to those reported previously for *P. infestans*, other oomycetes and in the fungus *Cylindrocarpon destructans* [10,46]. Based on these findings, we selected a fixed concentration of 25 μM to screen compounds for their ability to inhibit *P. infestans* growth (in order to identify molecules that are more effective than Mancozeb), with 1 mM cinnamaldehyde and 100 μM Mancozeb in positive control experiments.

### 2.2. Cinnamaldehyde and Aryl Aldehyde Screen

Due to the commercial availability of cinnamaldehyde based starting materials, *trans*-cinnamaldehyde (**7**), 4-fluorocinnamaldehyde **17** and α-methylcinnamaldehyde **18** were chosen as starting points for analogue formation and for testing against *P. infestans* (Figure 7). In addition, a series of aromatic aldehydes were identified for screening to probe the relevance of the conjugated aldehyde as a key functional group responsible for the effects (**20**–**24**). Indole-3-aldehydes **20** and **21** are structurally related to the tryptophan-derived plant metabolite parent (**19**) which has been identified to confer immunity in plants to microbial pathogen infection [47].

#### 2.2.1. Synthesis of Extended Cinnamaldehyde Derivatives

To probe the effect of the aldehyde group of cinnamaldehyde on the inhibition of *P. infestans* growth, *trans-*cinnamaldehyde (**7**) was treated with (carbethoxymethylene)triphenylphosphorane **25** (Scheme 1) yielding product as a colourless oil **26**. The same procedure was followed for the 4-fluorocinnamaldehyde **17** starting material however after 2 h there was still evidence of starting material. An additional 0.1 equivalent of the ylide **25** was added and stirred at room temperature for an additional 16 h. The crude material was isolated with the product **27** isolated as a colourless oil which solidified to colourless crystals overnight. 

α-Methylcinnamaldehyde **18** was also used in this reaction however it proved more problematic. Treatment with **25** and stirring in dichloromethane for 48 h showed no consumption of starting material by TLC analysis. A second equivalent of the ylide **25** was added and the reaction was let stir at 35 °C for a further 48 h. NMR analysis of the crude reaction mixture showed evidence of product formation but also the characteristic aldehyde starting material peak at 9.59 ppm with isolation of the pure product proving problematic.

#### 2.2.2. Synthesis of Cinnamaldehyde Thiosemicarbazone Derivatives

Given the structural distinction between cinnamaldehyde (**7**) and Mancozeb (**1**), it was envisaged that synthesis of a hybrid thiosemicarbazone derivative of the cinnamaldehyde could proffer improved activity based on the individual components. *trans*-Cinnamaldehyde (**7**) was dissolved in absolute ethanol and treated with thiosemicarbazide and a catalytic amount of acetic acid (Scheme 2). The starting material was consumed within 45 min and after isolation resulted in pure product **28** in 34% yield. The 4-fluorocinnamaldehyde reaction proceeded to completion after 16 h at 80 °C and the α-methyl derivative **30** was formed after 48 h at 80 °C. Both **28** and **29** crystallised as yellow needles whereas **30** was an amorphous white solid. 

### 2.3. Synthesis of Ellipticines and 9-Formylellipticines

The Gribble and Saulnier route to ellipticine (**15**) was chosen to probe the potential of ellipticines as it is a versatile and high yielding synthesis [48,49]. The amenable nature of the ellipticine scaffold allows for derivatisation at numerous positions (Scheme 3). The *N*-6 position of ellipticine was substituted by the addition of sodium hydride to deprotonate the indole nitrogen (Scheme 3). Due to different alkyl iodides possessing different reactivities, the time taken for the reaction to go to completion varied. For the methyl derivative **31**, 16 h was sufficient to achieve full conversion, whereas the ethyl derivative **32** took 72 h to go to completion and the isopropyl derivative **33** 48 h albeit with significantly reduced yield. The quantity of alkylating agent could only be used in slight excess as the addition of excess or extended reaction times can lead to quaternary salt formation on the pyridine nitrogen.

Ellipticine was then formylated at the nine-position to form the aldehyde by a method described by Plug et al. [50] and as first described by Duff (Scheme 3) [51,52]. This reaction was carried out on 6-substituted ellipticine derivatives to form 9-formylellipticine derivatives **34**, **35** and **36**. The formylation step was generally high yielding with good purity.

### 2.4. Screening of Cinnamaldehydes, Aromatic Aldehydes and Ellipticines against P. infestans

Compounds were tested for their ability to inhibit the mycelial growth and the production of zoospores by *P. infestans* [42]. The cytotoxicity of compounds against human cells was also evaluated, in the human embryonic kidney 293T (HEK-293T) cell line, using a sodium 3’-[1-(phenylaminocarbonyl)-3,4-tetrazolium]-bis (4-methoxy-6-nitro) benzene sulfonic acid hydrate (XTT) reduction assay [53,54]. For all assays, a fixed concentration of 25 μM was selected for screening purposes. As described in Section 2.1, Mancozeb (**1**) and cinnamaldehyde (**7**) were used in positive control experiments at their MIC values, of 100 μM and 1 mM, respectively. The vehicle for these compounds, DMSO, was used at a final concentration of 0.1% in negative control experiments.

#### 2.4.1. Phytophthora infestans Mycelial Growth Assay

In total, 17 compounds were evaluated for relative inhibitory effects on *P. infestans* mycelial growth at a concentration 25 µM after 5, 9 and 13 days.

##### Day 5

Cinnamaldehyde (**7**) and 9-formyl-6-methylellipticine **34** exhibit the best inhibition of *P. infestans* with no detectable growth for both compounds (Figure 8). All cinnamaldehyde derivatives **17**, **18**, **26**–**30** prove not as potent as the parent cinnamaldehyde (**7**), although this is tested at a much higher concentration. It is interesting that thiosemicarbazone cinnamaldehyde derivatives **28**–**30** show the slowest mycelial growth rate of all the test cinnamaldehyde series. As an example, the α-methyl thiosemicarbazone derivative **30** demonstrates only 38% growth which suggests the cinnamaldehyde/Mancozeb hybrid approach warrants further exploration. Unexpectedly, derivative **26** (ethyl-(2*E*)-5-phenylpenta-2,4-dienoate) caused a small but significant increase in *P. infestans* mycelial growth, of 113% relative to the vehicle control.

As expected, cinnamaldehyde (**7**) shows good inhibition of *P. infestans* which given its simple structure has been postulated to arise from the presence of an α,β-unsaturated aldehyde group and its potential to interact with cysteines in target proteins. Compounds (**7**), **17**, **18**, **20**–**24** and **34**–**36** arise from different chemical classes but all contain an aldehyde and in comparing their activity, there is no clear relationship between this functional group and mycelial growth.

The functionalised cinnamaldehydes **17** and **18** which contain the α,β-unsaturated aldehyde are identified with relatively low mycelial growth inhibition within this screen. Each of the proprietary aromatic aldehydes tested (**20**–**24**) show moderate to low inhibition of *P. infestans* with 5-bromoindole-3-carbaldehyde (**21**) seen as the most promising.

The most effective compounds in the screen are cinnamaldehyde (**7**) and 9-formyl-6-methylellipticine **34** and structurally these comprise an α,β-unsaturated aldehyde and an aromatic aldehyde. While it is evidently important for potency, this suggests that the aldehyde group is not solely responsible for inhibition of mycelial growth and that other factors must be explored.

Of the ellipticines, all six-alkylated compounds **31**, **34, 35** and **36** exerted a significant growth inhibitory effect, with **34** having a *C*-9 aldehyde installed on the A-ring which may contribute to the activity of the compound. Although Mancozeb (**1**) is four times the concentration of the remaining test compounds, it causes a relatively low inhibition of growth, with six other test compounds demonstrating more potent effects. 

##### Day 9

After nine days, it can be seen once again that cinnamaldehyde (**7**) and ellipticine aldehyde derivative **34** still demonstrate 100% inhibition of mycelial growth (Figure 9). Mancozeb (**1**) shows a decrease of 13% in the inhibition of mycelial growth since Day 5. The growth-promoting effect of compound **26** is maintained, at 6% greater than the vehicle control. Again, of particular note are the ellipticines **31**, **34** and **35** which show promising potency at this extended stage.

##### Day 13

The data in Figure 10 demonstrate that there is a high rate of mycelial growth in *P. infestans* incubated with many of the test compounds after 13 days. However, there is still a consistent undetectable growth for cinnamaldehyde (**7**) and only 6% growth for **34**. Mancozeb (**1**) has an increased rate of mycelium growth of 14% since Day 9. Aside from the ellipticines **31**, **34**–**36** the consistent activity of cinnamaldehyde/Mancozeb hybrid **30** is of note. The growth-promoting effect of **26** was not statistically significant after 13 days (the same as V, 100% growth).

#### 2.4.2. *P. infestans* Zoosporogenesis and Sporangiogenesis

The test compounds were investigated for their effects on zoosporogenesis. The number of zoospores produced after two weeks’ culture under each condition was counted using a haemocytometer. All compounds with the exception of the aldehydes **20**, **21**, **23** and **24**, and ellipticine (**15**), significantly reduced the production of zoospores. Several compounds demonstrated complete inhibition of zoospore formation with the ellipticines **31**, **34** and **35** identified as comparable with cinnamaldehyde (Figure 11). 

Decreased zoosporogenesis could result from decreased production of the sporangia, or from the defective generation of zoospores within these fruiting bodies. To test this, sporangiogenesis was measured in *P. infestans* cultured in the presence of aldehyde or cinnamaldehyde test compounds for a period of two weeks (Figure 12). Both Mancozeb, aldehyde **22** and ellipticine derivative **36** all caused significant reductions in sporangiogenesis. These fruiting structures were undetectable in *P. infestans* cultured in the presence of 1 mM cinnamaldehyde or 25 μM of ellipticines **31**, **34** and **35**, despite the latter three compounds permitting some mycelial growth after 13 days (see Figure 10). The parent compound ellipticine (**15**) caused a significant increase in sporangiogenesis, despite having no significant effects on mycelial growth or zoosporogenesis (Figure 10 and Figure 11).

#### 2.4.3. Human Embryonic Kidney Cell XTT Assay

To determine the toxicity of our screening compounds towards mammalian cells, an XTT reduction assay was performed. This assay is an improved form of the MTT (3-(4,5-dimethylthiazol-2-yl)-2,5-diphenyltetrazolium bromide) method, in which XTT is reduced to an orange formazan dye by mitochondrial dehydrogenases [53,54]. The levels of this product in the extracellular medium can be measured using spectrophotometry. The activity of mitochondrial dehydrogenases is partially determined by the number of metabolically active cells in the assay, and so XTT reduction can be used as an indirect indicator of cell viability. This XTT cytotoxicity assay was applied to human embryonic kidney 293T cells (HEK-293T) to determine the toxicity of the compounds against a cell type of mammalian origin. In total, seven cinnamaldehyde, five ellipticines in addition to five stock aldehydes were tested in triplicate at 25 µM (Figure 13) and incubated for 24 h. Cinnamaldehyde and Mancozeb were tested at concentrations of 1 mM and 100 μM, respectively.

##### Results from XTT Bioassay

There is a large variance in toxicity values throughout the library of compounds tested in this study (Figure 13). Given the electrophilic nature of the screening panel of compounds and potential for bioadduct formation, the establishment of relative toxicity is an important step in the development towards lead compound choice and potential for field trials. The most toxic compound with 5% XTT reduction relative to the vehicle control, is cinnamaldehyde (**7**). Mancozeb (**1**) shows high toxicity at 12% of control. There is a variance between the cinnamaldehyde derivatives with both α, β-unsaturated esters **26** and **27** demonstrating the lowest toxicity. Of the aldehydes, compound **23** displays the greatest inhibition of XTT reduction. All of the ellipticines, with the exception of compound **36**, display significant toxicity to this human cell line.

## 3. Discussion

Considering data from Day 5 mycelial growth inhibition and the XTT assay, relationships can be drawn from the data relating the compound structure to the inhibitory effects against *P. infestans.* From the outset, cinnamaldehyde (**7**) proved very effective at limiting *P. infestans* growth albeit at elevated concentrations while interestingly the standard Mancozeb (**1**) was much less effective across the evaluations. The synthetic modifications of cinnamaldehyde (**17**–**18**, **26**–**30**) proved instructive with overall less inhibitory ability than the parent (**7**). In light of the relative concentrations used (1 mM versus 25 μM), the promising data for **28** and **30** should be explored further especially given their limited cytotoxicity against human cells. Both compounds arise from the thiosemicarbazone cinnamaldehyde Mancozeb hybrid and have significant growth inhibition of *P. infestans* (56% and 38%, respectively) so it is evident that aldehyde conversion can have an effect on growth. Substitution on the aromatic ring, at the two-position in the chain or extension to a conjugated ester, appears to have little beneficial effect. In relation to the toxicities of the cinnamaldehyde thiosemicarbazone derivatives, 4-fluoro **29** proves to be the most toxic at 65% cell viability. In addition, 4-fluorocinnamaldehyde **17** is also more toxic than α-methylcinnamaldehyde **18** with 47% viability in comparison to 96%. Conversion of 4-fluorocinnamaldehyde **17** to the extended unsaturated ester **27** increases the cell viability by 53% indicating that the aldehyde induces a greater cytotoxic effect. It is possible that the 4-fluoro derivatives have higher cytotoxic effects due to the electron-withdrawing influence the fluorine atom has on the Michael acceptor. 

The commercial aromatic aldehydes had little effect at limiting *P. infestans* growth although the 5-bromoindole-3-carbaldehyde **21** stands out at 52% growth versus control. This needs to be balanced against the 15% decrease in XTT reduction which limits the further use of this specific framework. Indole derivatives and indole-3-carbaldehydes are natural products derived from the amino acid tryptophan and have been shown to provide innate plant resistance against pathogens so this deserves consideration for future investigations [47]. It is also of interest to note that the indole is a component of the subsequent ellipticines screened. 

On Day 5, the parent compound ellipticine (**15**) had little effect on mycelial growth, 106% relative to the control. When this compound is methylated at the *N-6* position to generate **31**, the growth rate drops to 21%, which decreases to no detectable growth when **31** is formylated at the nine-position to form **34**. It appears that not only is substituting the *N*-6 site important, the alkyl group that is substituted also plays a role. 9-Formyl-6-methylellipticine **34** shows 0% growth of *P. infestans* whereas increasing to 6-ethyl **35** gives 69% growth and 6-isopropyl **36** demonstrates a 56% growth. Evidently, the size of the substituent at the *N*-6 position impacts the potency of ellipticine derivatives [52,55]. It is postulated that protecting the *N*-6 position plays a vital role in lowering the growth rate of *P. infestans* and the addition of an aldehyde group at the *C*-9 position results in complete inhibition of growth. Compound **34** is as effective as cinnamaldehyde (**7**) at a dosage 40 times less and suggests that there is real merit in further exploration of this structure. On extension of the assay to Day 9 and Day 13, a consistent pattern emerges and only after Day 13 do we record any growth with the ellipticine **34** (6%). 

Evaluation of zoosporogenesis identified that compounds **31**, **34**, **35** and cinnamaldehyde (**7**) completely eradicated the formation of zoospores which is an essential mechanism of growth. This is the first time an ellipticine has recorded such activity.

There is no strong structural correlation between toxicity toward a human cell line and inhibition of *P. infestans* mycelial derived from this study (Figure 14). The commercial pesticide Mancozeb (**1**) expressed a low cell viability value of 12% and a midrange level of *P. infestans* inhibition at 50% even at four times the concentration of the other test compounds. From the data collected, Mancozeb (**1**) does not represent the strongest candidate in this study. In addition, cinnamaldehyde (**7**) is at a greater concentration than the test compounds and has also been shown by Hu et al. [42] to have an effective concentration of 2 mM and therefore, also appears to have limitations. The viability percentage for cinnamaldehyde of 5% represents a serious concern for its potential use as a fungicidal agent. With respect to the cinnamaldehyde derivatives, all are less cytotoxic than cinnamaldehyde (**7**) (albeit at lower concentrations) and of special note is compound **30** (methyl-substituted cinnamaldehyde thiosemicarbazone) with excellent cytotoxicity to mycelial growth inhibition profile worthy of further investigation. 

The parent compound ellipticine (**15**) has a low cell viability of 21%. Substitution at the *N*-6 position, for example, 6-methylellipticine **31** lowers the toxicity to 37% which lowers further to 40% upon addition of a formyl group at the nine-position to form 9-formyl-6-methylellipticine **34**. In addition to substitution playing a vital role in toxicity, the size of the substituted group also has an effect. In comparison to 9-formyl-6-methylellipticine **34**, 9-formyl-6-ethylellipticine **35** and 9-formyl-6-isopropylellipticine **36** show cell viabilities of 14% and 82%, respectively. It is evident that the isopropyl substituent expresses the highest cell viability for these compounds. Of the ellipticines, **34** provides the best cytotoxicity to *P. infestans* growth inhibition profile (Figure 14) and merits further exploration of its activity. It is significant to note that the screening concentration of 25 μM is most likely not the optimal concentration for *Phytophthora* growth inhibition and hence this profile has the potential to be significantly improved.

## 4. Materials and Methods

Solvents were distilled prior to use by the following methods: ethyl acetate was distilled from potassium carbonate; THF was freshly distilled from sodium and benzophenone and hexane was distilled prior to use. Organic phases were dried using anhydrous magnesium sulphate. Where room temperature is quoted in a method, this is within the temperature range 15–20 °C.

All commercial reagents were used without further purification unless otherwise stated. Alkyllithium reagents were titrated prior to use using the Gilman double titration procedure as follows; Two 100 mL conical flasks were prepared, the first one containing water (25 mL) and the second one containing dibromoethane (2 mL, stoppered with a SubaSeal under nitrogen with a provision for pressure release). The alkyllithium reagent (1 mL) was added via syringe to each flask [56,57]. The reaction with dibromoethane was more vigorous than that with water, and the flask was swirled during addition. Two drops of phenolphthalein were added to both flasks. The first flask was titrated against 0.1 M HCl until the purple colour disappeared. The SubaSeal was removed from the second flask and water (25 mL) was added, forming a biphasic mixture and was titrated as before. (Note: constant stirring was maintained to ensure mixing of both phases during titration). Titre value 1 represents the total base (alkyllithium and inorganic base) while titre 2 represents the free base (inorganic base not resulting from the alkyllithium). Thus, the molarity of the alkyllithium reagent was calculated using the following equation: molarity of alkyllithium (M) = [(titre 1−titre 2) × 0.1].

In reactions utilising alkyllithium and sodium hydride, all glassware was flame dried under nitrogen prior to use. All low-temperature reactions were carried out in a three-necked round-bottomed flask equipped with a low-temperature thermometer and the internal temperatures are quoted. Syringes were used to transfer small volumes of alkyllithium reagents while cannulation from the reagent bottle into a precalibrated addition funnel was used for larger volumes (>15 mL). Low-temperature reactions used the following cooling mixture: −100 °C, absolute ethanol and liquid nitrogen. 

^1^H (300 MHz) and ^13^C (75.5 MHz) NMR spectra were recorded on a Bruker AVANCE 300 NMR spectrometer. ^1^H (600 MHz) and ^13^C (150.9 MHz) NMR spectra were recorded on a Bruker AVANCE III 600 NMR spectrometer equipped with a Bruker Dual C/H cryoprobe or a Bruker Broadband Observe H&F cryoprobe. All spectra were recorded at 300 K (26.9 °C) in deuterated dimethylsulfoxide (DMSO-*d*_6_) using DMSO-*d*_6_ as the reference peak or in deuterated chloroform (CDCl_3_) using trimethylsilane as an internal standard unless otherwise specified. Chemical shifts (δ_H_ and δ_C_) are reported in parts per million (ppm) relative to the reference peak. Coupling constants (*J*) are expressed in Hertz (Hz). Splitting patterns in ^1^H spectra are designated as s (singlet), br s (broad singlet), d (doublet), t (triplet), br t (broad triplet), q (quartet), sept (septet), dd (doublet of doublets), ddd (doublet of doublet of doublets) and m (multiplet). Signal assignments were supported by COSY (correlation spectroscopy) or HMBC (Heteronuclear Multiple-Bond Correlation spectroscopy) experiments where necessary. ^13^C NMR spectra were assigned (aromatic C, CH, CH_2_, CH_3_) with the aid of DEPT (Distortionless Enhancement by Polarisation Transfer) experiments run in DEPT-90, DEPT-135 and DEPT-q modes. Specific assignments were made using HSQC (Heteronuclear Single Quantum Correlation) and HMBC (Heteronuclear Multiple-Bond Correlation) experiments. All spectroscopic data for known compounds were in agreement with those previously reported unless otherwise stated. Samples used for comparison (stacked plots) were run at equal concentrations (6–8 mg per 0.65 mL solvent). Carbon analysis is given for compounds that are novel or where full analysis has not been published in the literature. 

Infrared spectra were recorded on a Bruker Tensor 37 FT-IR spectrophotometer interfaced with Opus version 7.2.139.1294 over a range of 400–4000 cm^−1^. An average of 16 scans was taken for each spectrum obtained with a resolution of 4 cm^−1^. Melting points were measured on a Uni-Melt Thomas Hoover capillary melting point apparatus and are uncorrected. Melting points or boiling points were not obtained for semi-solids or oils. 

Thin Layer Chromatography (TLC) was carried out on precoated silica gel plates (Merck 60 F_254_). Visualisation was achieved by UV light detection (254 nm or 366 nm), Wet flash chromatography was carried out using Kieselgel silica gel 60, 0.040–0.063 mm (Merck). 

Low-resolution mass spectra were recorded on a Waters Quattro Micro triple quadrupole spectrometer (QAA1202) in electron spray ionisation mode (ESI) using acetonitrile/water (1:1) containing 0.1% Formic acid as eluent. High-resolution mass spectrometry (HRMS) spectra were recorded on Waters Vion IMS (model no. SAA055K) in electron spray ionisation mode (ESI) using acetonitrile/water (1:1) containing 0.1% Formic acid as eluent. 

Ethyl-(2*E*)-5-phenylpenta-2,4-dienoate **26**

*trans-*Cinnamaldehyde **7** (0.100 g, 0.76 mmol) in dichloromethane (5 mL) was treated with (carbethoxymethylene)triphenylphosphorane **25** (0.267 g, 0.76 mmol) and allowed stir at room temperature for 1 h. The solvent was removed under reduced pressure and hexane was added. The triphenylphosphine oxide byproduct precipitated out as a pale pink solid and was isolated by vacuum filtration. The mother liquor was transferred into a round-bottomed flask and solvent was removed under reduced pressure. The crude product was purified by column chromatography eluting in hexane: ethyl acetate (100:0–99:1). The product eluted as a colourless oil (0.128 g, 83.5%). IR ν_max_/cm^−1^: 3062, 3019, 2982, 1723, 1178; δ_H_ (300 MHz, CDCl_3_): 1.31 (3H, t, *J* 7.1, CH_2_C**H**_3_), 4.22 (2H, q, *J* 7.1, C**H**_2_CH_3_), 5.98 (1H, d, *J* 15.2, H-2), 6.80-6.94 (2H, m, H-5, H-4), 7.24–7.50 (6H, m, H-3, Ar**H**); *m*/*z* (ESI^+^) 202 [(M + H)^+^, 50%].

Ethyl-(2*E*,4*E*)-5-(4-fluorophenyl)penta-2,4-dienoate **27**

4-Fuorocinnamaldehyde **17** (0.100 g, 0.33 mmol) in dichloromethane (5 mL) was treated with (carbethoxymethylene)triphenylphosphorane **25** (0.115 g, 0.33 mmol) and allowed stir at room temperature for 2 h. Analysis by TLC showed residual starting material so an additional portion of (carbethoxymethylene)triphenylphosphorane (0.010 mg, 0.03 mmol) was added and allowed stir overnight. The solvent was removed under reduced pressure and hexane was added. The white triphenylphosphine oxide byproduct precipitated out and was isolated by vacuum filtration. The mother liquor was transferred into a round-bottomed flask and solvent was removed under reduced pressure. The crude product was purified by column chromatography eluting in hexane: ethyl acetate (100:0–97:3). The product eluted as a colourless oil but solidified overnight to a white crystalline solid. (0.43 g, 64.5%). m.p. 41-43 °C (Lit. m.p. 44–46 °C) [58]; IR ν_max_/cm^−1^: 2986, 1696, 1238, 1176; δ_H_ (300 MHz, CDCl_3_): 1.31 (3H, t, *J* 6.9, CH_2_C**H**_3_), 4.23 (2H, q, *J* 6.9, C**H**_2_CH_3_), 5.98 (1H, d, *J* 15.3, H-2), 6.77 (1H, dd, *J* 15.3, 9.9, H-4), 6.86 (1H, d, *J* 15.3, H-5), 7.04 (2H, t, *J* 8.6, H-6′, H-2′), 7.36-7.47 (3H, m, H-5′, H-3′, H-3); *m*/*z* (ESI^+^) 221 [(M + H)^+^, 30%].

2-((*E*)-3-Phenylallylidene)hydrazine-1-carbothioamide **28**

To a solution of *trans-*cinnamaldehyde **7** (0.100 g, 0.76 mmol) in absolute ethanol (5 mL) was added thiosemicarbazide (0.069 mg, 0.76 mmol) and acetic acid (2 drops). The reaction mixture was heated to 80 °C and allowed stir for 45 min. The reaction mixture was cooled to 0 °C for 3 h without agitation. A precipitate formed and was isolated by vacuum filtration. The crude product was recrystallised from ethyl acetate: hexane to form pale yellow crystals (54 mg, 34.3%) m.p. 115–117 °C (Lit. m.p. 110–113 °C) [59]; IR ν_max_/cm^−1^: 3413, 3255, 3155, 3028, 2981, 1610, 1590, 1536, 1372, 1291, 818; δ_H_ (300 MHz, CDCl_3_): 6.47 (1H, br s, N**H**_2_) 6.78 (1H, dd, *J* 16.0, 9.0, H-2), 6.95 (1H, d, *J* 16.0, H-3), 7.15 (1H, br s, N**H**_2_), 7.30–7.40 (3H, m, H-2′, H-3′, H-4′), 7.45 (2H, dd, *J* 8.1, 2.0, H-1′, H-5′), 7.75 (1H, d, *J* 9.0, H-1), 10.08 (1H, s, CNN**H**); *m*/*z* (ESI^+^) 206 ((M + H)^+^, 100%].

2-((*E*)-3-(4-Fluorophenyl)allylidene)hydrazine-1-carbothioamide **29**

To a solution of 4-fluorocinnamaldehyde **17** (0.100 g, 0.67 mmol) in absolute ethanol (5 mL) was added thiosemicarbazide (0.060 g, 0.67 mmol) and acetic acid (2 drops). The reaction mixture was heated to 80 °C and allowed stir for 16 h. The reaction mixture was cooled to 0 °C for 3 h without agitation. A precipitate formed and was isolated by vacuum filtration. The crude product was recrystallised from ethyl acetate: hexane to form pure, pale yellow crystals (0.89 g, 59.7%) m.p. 144–146 °C; IR ν_max_/cm^−1^: 3386, 3249, 3149, 2985, 1610, 1600, 1528, 1505, 1475, 1286, 1156, 1082, 814; δ_H_ (300 MHz, CDCl_3_): 6.40 (1H, br s, N**H**_2_), 6.72 (1H, dd, *J* 16.0, 9.0, H-3), 6.90 (1H, d, *J* 16.0, H-2), 7.05 (2H, t, *J* 8.6, H-2′, H-6′), 7.13 (1H, br s, N**H**_2_), 7.43 (2H, dd, *J* 8.6, 5.4, H-5′, H-3′), 7.72 (1H, d, *J* 9.0, H-1), 9.95 (1H, br s, CNN**H**); δ_C_ (75.5 MHz, DMSO-*d*_6_): 79.63 (CH), 116.27 (CH, *J*_C-F_ 21.7, aromatic CH), 125.41 (CH, aromatic CH), 125.44 (CH, aromatic CH), 129.47 (d, CH *J*_C-F_ 8.6, aromatic CH), 133.02 (d, C, *J*_C-F_ 3.3, aromatic C), 138.09 (CH), 145.11 (CH), 162.75 (d, C, *J*_C-F_ 247.1, CF), 178.19 (C, **C**S); *m*/*z* (ESI^+^) 224 [(M + H)^+^, 100%]; HRMS (ESI^+^): Exact mass calculated for [C_10_H_11_FN_3_S]^+^ 224.0652. Found 224.0651.

2-Methyl-3-phenylallylidene hydrazine-1-carbothioamide **30**

To a solution of α-methylcinnamaldehyde **18** (0.100 g, 0.68 mmol) in absolute ethanol (5 mL) was added thiosemicarbazide (0.062 g, 0.68 mmol) and acetic acid (2 drops). The reaction mixture was heated to 80 °C and allowed stir for 48 h. The reaction mixture was cooled to 0 °C for 3 h without agitation. A precipitate formed and was isolated by vacuum filtration. The crude product was recrystallised from ethyl acetate: hexane to form a paper-like white solid (0.76 g, 51.0%) m.p. 173–175 °C (Lit. m.p. 170–172 °C) [60]; IR ν_max_/cm^−1^: 3423, 3241, 3149, 2965, 1602, 1508, 819; δ_H_ (300 MHz, CDCl_3_): 2.12 (3H, s, CH_3_-2), 6.39 (1H, br s, N**H**_2_), 6.80 (1H, s, H-3), 7.14 (1H, br s, N**H**_2_), 7.27-7.41 (5H, m, Ar**H**), 7.69 (1H, s, H-1), 9.73 (1H, br s, CNN**H**); *m*/*z* (ESI^+^) 220 [(M + H)^+^, 100%].

5,6,11-Trimethyl-6*H*-pyrido [4,3-*b*]carbazole **31**

5,11-Dimethyl-6*H*-pyrido[4,3*b*]carbazole **15** (1.50 g, 6.10 mmol) in anhydrous DMF (20 mL) was treated with NaH (0.375 g, 60% dispersion in mineral oil 9.4 mmol,) at 0 °C under nitrogen and allowed stir for 30 min, forming a deep red suspension. Iodomethane (0.39 mL, 6.20 mmol) was added and the reaction was allowed warm to room temperature and stir overnight. The reaction mixture was cooled to 0 °C and water (100 mL) was added producing a pale orange precipitate. The yellow product was isolated by vacuum filtration and washed with water (150 mL) and did not require further purification (1.46 g, 91.8%). m.p 205–206 °C (Lit. m.p 202–204 °C) [61]; IR ν_max_/cm^−1^: 2921, 2852, 1588, 1476, 1449, 1241, 1102, 806; δ_H_ (300 MHz, DMSO-*d*_6_): 3.03 (3H, s, CH_3_-5), 3.19 (3H, s, CH_3_-11), 4.13 (3H, s, NCH_3_), 7.31 (1H, overlapping ddd, *J* 8.0, 6.4, 1.5, H-9), 7.58 (1H, dd, *J* 8.0, 1.5, H-8), 7.62 (1H, d, *J* 1.5, H-7), 7.99 (1H, d, *J* 6.0, H-4), 8.37 (1H, d, *J* 8.0, H-10), 8.45 (1H, d, *J* 6.0, H-3), 9.68 (1H, s, H-1); *m*/*z* (ESI^+^) 261 [(M + H)^+^, 100%].

6-Ethyl-5,11-dimethyl-6*H*-pyrido[4,3-*b*]carbazole **32**

5,11-Dimethyl-6*H*-pyrido[4,3-*b*]carbazole **15** (3.33 g, 13.5 mmol) in anhydrous DMF (60 mL), at 0 °C under nitrogen, was treated with sodium hydride (0.812 g, 20.3 mmol, 60% dispersion in mineral oil) and left to stir for 30 min forming a red suspension. Iodoethane (1.14 mL, 2.22 g, 14.2 mmol) was added and the reaction was allowed to warm to room temperature and was left to stir for a total of 72 h. The reaction mixture was cooled on ice and water (200 mL) was added. The aqueous layer was extracted with dichloromethane-methanol 90:10 (5 × 50 mL). The combined organic extracts were washed with water (5 × 50 mL) and brine (1 × 50 mL), dried over magnesium sulphate, filtered and the solvent removed under reduced pressure. Purification by column chromatography eluting with dichloromethane-methanol (99:1) gave the product as a yellow solid (2.35 g, 63.4%). m.p. 164–165 °C; ν_max_/cm^−1^ (KBr): 3020, 2966, 2926, 1596, 1588, 1475, 1395, 1382, 1346, 1232, 741; δ_H_ (600 MHz, DMSO-*d*_6_): 1.33 [3H, t, *J* 7.1, N(6)CH_2_CH_3_], 2.84 [3H, s, C(5)CH_3_], 3.03 [3H, s, C(11)CH_3_], 4.50 [2H, q, *J* 7.1, N(6)CH_2_CH_3_], 7.26 [1H, t, *J* 7.4, C(9)H], 7.53–7.58 [2H, m, C(7)H, C(8)H], 7.92 [1H, d, *J* 6.1, C(4)H], 8.26 [1H, d, *J* 7.8, C(10)H], 8.41 [1H, d, *J* 6.0, C(3)H], 9.58 [1H, s, C(1)H]; δ_C_ (150.9 MHz, DMSO-*d*_6_): 13.4 [CH_3_, C(5)CH_3_], 14.7 [CH_3_, C(11)CH_3_], 15.4 [CH_3_, N(6)CH_2_CH_3_], 40.2 [CH_2_, N(6)CH_2_CH_3_], 108.7 (C, aromatic C), 109.4 (CH, aromatic CH), 116.3 (CH, aromatic CH), 120.1 (CH, aromatic CH), 122.4 (C, aromatic C), 123.3 (C, aromatic C), 124.2 (CH, aromatic CH), 124.5 (C, aromatic C), 127.7 (CH, aromatic CH), 128.8 (C, aromatic C), 134.0 (C, aromatic C), 140.3 (C, aromatic C), 141.2 (CH, aromatic CH), 144.1 (C, aromatic C), 149.9 (CH, aromatic CH); *m*/*z* (ESI^+^): 275 [(M + H)^+^ 100%]; HRMS (ESI^+^): Exact mass calculated for C_19_H_19_N_2_^+^ 275.1548. Found 275.1547.

6-Isopropyl-5,11-dimethyl-6*H*-pyrido[4,3-*b*]carbazole **33**

5,11-Dimethyl-6*H*-pyrido[4,3-*b*]carbazole **15** (2.31 g, 9.37 mmol) was added portion-wise to a suspension of sodium hydride (0.75 g, 60% dispersion in mineral oil, equivalent to 0.45 g sodium hydride, 18.8 mmol) in dimethylformamide (10 mL) forming a deep red suspension which was stirred at room temperature for 45 min under nitrogen. 2-Iodopropane (1.10 mL, 1.90 g, 11.2 mmol) in dimethylformamide (5 mL) was added dropwise and the reaction was stirred for 48 h. The reaction mixture was cooled on ice and water (50 mL) added. The aqueous layer was extracted with dichloromethane—Methanol 90:10 (3 × 50 mL). A precipitate in the aqueous layer which would not extract was filtered and found to be unreacted ellipticine (402 mg). Combined organic layers were washed with water (5 × 20 mL) and brine (1 × 30 mL), dried over magnesium sulphate and solvent removed under reduced pressure. Purification by column chromatography eluting with dichloromethane – methanol 99:1 gave the product as a yellow solid (0.96 g, 35.5%). m.p. 168–170 °C; ν_max_/cm^−1^ (KBr): 3435, 3079, 2970, 2933, 1598, 1587, 1469, 1390, 1367, 1345, 1288, 1236, 1104, 804, 741; δ_H_ (600 MHz, DMSO-*d*_6_): 1.59 [6H, d, *J* 6.9, N(6)CH(CH_3_)_2_], 2.87 [3H, s, C(5)CH_3_], 3.15 [3H, s, C(11)CH_3_], 5.32 [1H, sept, *J* 7.0, N(6)CH(CH_3_)_2_], 7.28 [1H, overlapping ddd, *J* 7.8, 7.2, 0.5, C(9)H], 7.51 [1H, overlapping ddd, *J* 8.2, 7.2, 1.0, C(8)H], 7.77 [1H, d, *J* 8.2, C(7)H], 7.92 [1H, d, *J* 6.0, C(4)H], 8.34 [1H, d, *J* 7.8, C(10)H], 8.43 [1H, d, *J* 5.8, C(3)H], 9.66 [1H, s, C(1)H]; δ_H_ (600 MHz, DMSO-*d*_6_): 14.8 [CH_3_, C(11)CH_3_], 15.3 [CH_3_, C(5)CH_3_], 21.4 [2 × CH_3_, N(6)CH(CH_3_)_2_], 49.6 [CH, N(6)CH(CH_3_)_2_], 109.6 (C, aromatic C), 113.6 (CH, aromatic CH), 116.6 (CH, aromatic CH), 120.2 (CH, aromatic CH), 122.9 (C, aromatic C), 124.5 (CH, aromatic CH), 124.9 (C, aromatic C), 125.3 (C, aromatic C), 127.0 (CH, aromatic CH), 128.3 (C, aromatic C), 134.6 (C, aromatic C), 141.4 (CH, aromatic CH), 143.1 (C, aromatic C), 143.2 (C, aromatic C), 149.9 (CH, aromatic CH); *m*/*z* (ESI^+^): 289 [(M + H)^+^, 100%]; HRMS (ESI^+^): Exact mass calculated for C_20_H_21_N_2_^+^ 289.1705. Found 289.1707.

5,6,11-Trimethyl-6*H*-pyrido[4,3-*b*]carbazole-9-carbaldehyde **34**

5,6,11-Trimethyl-6*H*-pyrido[4,3-*b*]carbazole **31** (0.600 g, 2.33 mmol) was dissolved in trifluoroacetic acid (30 mL) and hexamethylenetetramine (3.24 g, 23.41 mmol) added portion-wise. The reaction was heated to reflux for 1 h. Upon cooling to room temperature, the mixture was cooled to 0 °C and poured into water (200 mL). Solid sodium bicarbonate was added until neutralised and the mixture was extracted with dichloromethane-methanol (150 mL × 4, 9:1). The combined organic layers were washed with water (2 × 100 mL) and brine (2 × 100 mL), dried, filtered and solvent removed under reduced pressure to yield an orange solid (0.53 g, 81.0%). m.p. 219–222 °C (Lit. m.p. 222-223 °C) [55,62]; IR ν_max_/cm^−1^: 3214, 2922, 2852, 1674, 1586, 1461, 1380, 1356, 1099, 983; δ_H_ (300 MHz, DMSO-*d*_6_): 3.01 (3H, s, CH_3_-5), 3.21 (3H, s, CH_3_-11), 4.16 (3H, s, NCH_3_), 7.75 (1H, d, *J* 8.7, H-7), 8.01 (1H, d, *J* 5.8, H-4), 8.07 (1H, dd, *J* 8.7, 1.5, H-8), 8.49 (1H, d, *J* 5.8, H-3), 8.78 (1H, d, *J* 1.5, H-10), 9.70 (1H, s, H-1), 10.10 (1H, s, C**H**O); *m*/*z* (ESI^+^) 289 [(M + H)^+^, 100%].

6-Ethyl-5,11-dimethyl-6*H*-pyrido[4,3-*b*]carbazole-9-carbaldehyde **35**

6-Ethyl-5,11-dimethyl-6*H*-pyrido[4,3-*b*]carbazole **32** (1.00 g, 3.65 mmol) was stirred in trifluoroacetic acid (40 mL) and hexamethylenetetramine (5.11 g, 36.5 mmol) was added portion-wise. The reaction was heated to reflux for 40 min. Upon cooling the reaction mixture was cooled on ice and poured into water (250 mL). The reaction mixture was neutralised using solid sodium bicarbonate [Caution: vigorous reaction] and extracted with dichloromethane-methanol 90:10 (3 × 100 mL). The combined organic extracts were washed with water (2 × 100 mL) and brine (1 × 100 mL), dried over magnesium sulphate, filtered and the solvent removed under reduced pressure to give the product as an orange solid which was used without further purification (1.00 g, 91.0%). m.p. 192–196 °C; ν_max_/cm^−1^ (KBr): 2970, 2929, 2720, 1682, 1589, 1452, 1381, 1315, 1235, 1100, 809; δ_H_ (500 MHz, DMSO-*d*_6_): 1.42 [3H, t, *J* 6.9, N(6)CH_2_CH_3_], 2.95 [3H, s, C(5)CH_3_], 3.19 [3H, s, C(11)CH_3_], 4.66 [2H, q, *J* 6.8, N(6)CH_2_CH_3_], 7.78 [1H, d, *J* 8.5, C(7)H], 8.03 [1H, d, *J* 5.9, C(4)H], 8.08 [1H, d, *J* 8.4, C(8)H], 8.49 [1H, d, *J* 5.6, C(3)H], 8.77 [1H, s, C(10)H], 9.79 [1H, s, C(1)H], 10.10 [1H, s, C(9)CHO]; δ_C_ (125.8 MHz, DMSO-*d*_6_): 13.4 [CH_3_, C(5)CH_3_], 14.9 [CH_3_, C(11)CH_3_], 15.5 [CH_3_, N(6)CH_2_CH_3_], 40.7 [CH_2_, N(6)CH_2_CH_3_], 109.9 (CH, aromatic CH), 110.3 (C, aromatic C), 116.6 (CH, aromatic CH), 122.9 (C, aromatic C), 123.3 (C, aromatic C), 123.9 (C, aromatic C), 127.1 (CH, aromatic CH), 128.9 (CH, aromatic CH), 129.3 (C, aromatic C), 129.7 (C, aromatic C), 134.5 (C, aromatic C), 140.6 (C, aromatic C), 141.9 (CH, aromatic CH), 147.9 (C, aromatic C), 150.2 (CH, aromatic CH), 192.4 [C, C(9)CHO]; *m*/*z* (ESI^+^): 303 [(M + H)^+^, 100%]; HRMS (ESI^+^): Exact mass calculated for C_20_H_19_N_2_O^+^ 303.1497. Found 303.1485.

6-Isopropyl-5,11-dimethyl-6*H*-pyrido[4,3-*b*]carbazole-9-carbaldehyde **36**

6-Isopropyl-5,11-dimethyl-6*H*-pyrido[4,3-*b*]carbazole **33** (0.81 g, 2.82 mmol) was stirred in trifluoroacetic acid (40 mL) and hexamethylenetetramine (3.96 g, 28.2 mmol) was added portion-wise. The reaction was heated to reflux for 40 min. Upon cooling the reaction mixture was cooled on ice and poured into water (250 mL). The reaction mixture was neutralised using solid sodium bicarbonate [Caution: vigorous reaction] and extracted with dichloromethane-methanol 90:10 (3 × 100 mL). The combined organic extracts were washed with water (2 × 100 mL) and brine (1 × 100 mL), dried over magnesium sulphate, filtered and the solvent removed under reduced pressure to give the product as an orange solid which was used without further purification (0.836 g, 93.7%). m.p. 200–205 °C; ν_max_/cm^−1^ (KBr): 3412, 2969, 2929, 1683, 1588, 1574, 1386, 1321, 1286, 1240, 1192, 1140, 1104, 812; δ_H_ (600 MHz, DMSO-*d*_6_): 1.67 [6H, d, *J* 7.0, N(6)CH(CH_3_)_2_], 2.94 [3H, s, C(5)CH_3_], 3.24 [3H, s, C(11)CH_3_], 5.45 [1H, sept, *J* 6.9, N(6)CH(CH_3_)_2_], 7.97 [1H, d, *J* 8.6, C(7)H], 8.01 [1H, d, *J* 6.1, C(4)H], 8.04 [1H, dd, *J* 8.6, 1.4, C(8)H], 8.50 [1H, d, *J* 6.0, C(3)H], 8.84 [1H, d, *J* 1.1, C(10)H], 9.73 [1H, s, C(1)H], 10.11 [1H, s, C(9)CHO]; δ_C_ (150.9 MHz, DMSO-*d*_6_): 14.9 [CH_3_, C(11)CH_3_], 15.3 [CH_3_, C(5)CH_3_], 21.3 [2 × CH_3_, N(6)CH(CH_3_)_2_], 50.0 [CH, N(6)CH(CH_3_)_2_], 110.8 (C, aromatic C), 113.6 (CH, aromatic CH), 116.9 (CH, aromatic CH), 123.2 (C, aromatic C), 124.1 (C, aromatic C), 125.2 (C, aromatic C), 127.2 (CH, aromatic CH), 128.0 (CH, aromatic CH), 129.1 (C, aromatic C), 129.3 (C, aromatic C), 135.0 (C, aromatic C), 141.9 (CH, aromatic CH), 143.0 (C, aromatic C), 147.1 (C, aromatic C), 150.1 (CH, aromatic CH), 192.6 [C, C(9)CHO]; *m*/*z* (ESI^+^): 317 [(M + H)^+^, 100%]; HRMS (ESI^+^): Exact mass calculated for C_21_H_21_N_2_O^+^ 317.1654. Found 317.1651.

*Phytophthora infestans* Mycelial Growth Assays

*Phytophthora infestans* strain 88,069 (mating type A1) was originally obtained from Professor Sophien Kamoun (The Sainsbury Laboratory, Norwich, UK) and maintained on RyeA agar at 20 °C in the dark [63]. For mycelial growth assay, compounds were dissolved in DMSO in a vial and 15 mL Rye B agar added, at a temperature of approximately 50 °C (the final concentration of DMSO was 0.1%). The vial was shaken to distribute the contents and the mixture was then poured into a clean, sterile 10 cm Petri dish. A 10 mm plug of *P. infestans* mycelium was placed in the centre of the dish, which was then sealed with parafilm. The experiments were carried out in triplicate at 25 µM (unless otherwise stated), stored at 20 °C in the dark and checked periodically for mycelial growth.

*Phytophthora infestans* Zoosporogenesis and Sporangiogenesis Assays

These were performed essentially as described by Lu et al. [39]. Following two weeks of culture on Rye B agar, sporangia were collected from plates in 5 mL of modified Petri’s solution (MPS, 5 mM CaCl_2_, 1 mM MgSO_4_, 1 mM KH_2_PO_4_ and 0.8 mM KCl), by scraping with a scalpel blade and were transferred to a 50 mL sterile tube. Plates were washed with another 5 mL of MPS and this was combined with the first 5 mL. To initiate zoospore release, sporangia were place on ice, in the dark, for 3h. Sporangia or zoospores were counted by brightfield microscopy using a 10× objective and a haemocytometer, in duplicate.

XTT reduction assay

Human embryonic kidney-293T (HEK-293T) cells were obtained from LGC Standards (Middlesex, UK) and were maintained in Dulbecco’s Modified Eagle’s Medium containing 10% foetal bovine serum, 100 units/mL penicillin and 100 μg/mL streptomycin, at 37 °C in a humidified atmosphere of 5% CO_2_/95% air. For assays, cells were subcultured on 96-well microtitre plates at a density of 5 × 10^4^ cells/well, in a volume of 50 mL. Following overnight culture, 50 mL of each test compound at twice their final concentration (50 mM for all apart from cinnamaldehyde (2 mM) and Mancozeb (200 μM)) was added and incubated for another 24 h. Cells were incubated with 100 mL XTT reagents (Sigma-Aldrich, Ireland) for an additional 4 h, after which, the absorbance of the extracellular formazan reduction product was measured at a wavelength of of 490 nm, with a reference wavelength of 675 nm [53,54], using a microtitre plate spectrophotometer.

Statistical Methods

Numerical data were statistically compared using one-way analysis of variance (ANOVA) and Tukey’s post hoc test (using ezANOVA software, available at: https://people.cas.sc.edu/rorden/ezanova/index.html.) A probability threshold of *p* < 0.05 was taken as statistically significant. Generation of histograms and linear regression analyses were performed using Microsoft Excel.

## 5. Conclusions

It is evident from the synthesis and biological evaluation that both cell viability and inhibitory effects are impacted by discreet modifications of the cinnamaldehyde and ellipticine scaffolds. The mycelium growth assay provided information at Days 5, 9 and 13 of *P. infestans* incubation. From an initial focus on the cinnamaldehyde derivatives, the cinnamaldehyde Mancozeb hybrid proved fruitful and is a template for further discovery given its excellent *P. infestans* growth inhibition/toxicity profile. While the effects of cinnamaldehyde derivatives were somewhat predicted, uniquely in the screen of aldehydes the ellipticine aldehydes have emerged with excellent *Phytophthora* growth inhibitory properties. The aldehyde functionality on ellipticine has been shown to increase significantly the inhibitory effects of the compound and limit the toxicity to an extent though this needs to be tempered further before progressing. In contrast, the stock aldehydes did not demonstrate strong inhibition which suggests that there are additional factors affecting the oomycete growth.

The effects of ellipticine substitution have identified that alkylation of the six-position considerably improves mycelial growth inhibition. Ellipticines **31** and **34** have significant growth inhibitory properties on mycelia and inhibit zoosporogenesis and sporangiogenesis in comparison to ellipticine (**15**). It is evident from the biological data that **34** demonstrates the greatest inhibitory effects on *P. infestans* with a mycelium growth of just 6% after 13 days of incubation and the comparison with cinnamaldehyde is favourable given the respective dosages. The XTT cytotoxicity assay has highlighted that cinnamaldehyde and ellipticine compounds were amongst the most highly toxic of all the compounds tested and currently limits their potential use as fungicides. However, the discovery of 9-formyl-6-methylellipticine **34** as a potent growth inhibitor is the subject of our investigation into the influence of 2- and 6-alkylated 9-formylellipticines on *P. infestans* growth [64]. This study has resulted in promising new cinnamaldehyde and ellipticine candidates for *P. infestans* control that warrant further study to examine the role of substituents and mechanism of action.

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
