# Peer review of "Inhibitory Properties of Aldehydes and Related Compounds against Phytophthora infestans—Identification of a New Lead"

_pathogens, 2020, doi:10.3390/pathogens9070542_

Round 1

Reviewer 1 Report

Dear Authors,

now i have finished reading your manuscript and I was very happy to hear the things connected with this important topic.

I found just few minor things that should be improved before further processing.

Line 80: typo error in the name of species, should be P. capsici;

line 167: maybe you can consider to rewrite this aims paragraph into reported speech, with something like: "The aims of this study were to test the cinnamaldehyde and aromatic aldehyde dervates against P. infestans....

Line 707: strain instead of stain.

Also, maybe you can think about some figure with colony photos or sporangia treated with used compounds in the experiments, since this journal is offering the option to include color figure, but it is not essential for the presented work.

Good luck with your further research!

Author Response

The authors would like to thank the reviewer for their comments on our manuscript and have endeavoured to address each one in specific in the revised manuscript. The response to each specific comment is outlined below in red italics:

Reviewer 1

Dear Authors,

now i have finished reading your manuscript and I was very happy to hear the things connected with this important topic.

I found just few minor things that should be improved before further processing.

Line 80: typo error in the name of species, should be P. capsici; Amended as directed

line 167: maybe you can consider to rewrite this aims paragraph into reported speech, with something like: "The aims of this study were to test the cinnamaldehyde and aromatic aldehyde dervates against P. infestans.... Amended as directed

Line 707: strain instead of stain. Amended as directed

Also, maybe you can think about some figure with colony photos or sporangia treated with used compounds in the experiments, since this journal is offering the option to include color figure, but it is not essential for the presented work. Photos of mycelial growth have been added to the SI

Good luck with your further research! Thank you very much for your comments

Reviewer 2 Report

In general, the MS was well presented. The authors did successfully carried out the experiment and came out with some significant and interesting results. The literature cited was updated. However, they were a lit inconsistent in using the past tense in methods and results. They should all be in past tense. Further, there are some minor points that can be improved:

line 71 add citation reference no. after the authors.

line 84 place "plant genus" before "Cinnamona"

line 88 add citation no. after "Peligot"

line 85 add citation no. after "Quintanilla"

line 99 Insert "comma" after " formation"; delete "new"

line 101 delete (pathogenic skin fungi), readers should know the term "dermatophytes"

line 120 & 121. too much duplication. Either use "sporogenesis and zoosporogenesis" or simply " mycelial growth, sporangial formation and zoosporpore production as well as germination"

line 193 delete "hyphae"; "mycelia" alone will do

line 196 delete "of"

line 197 add "comma" after "incubation"

line 202 change "dinstinct" to "different"

line 234. add "period" after "material" and start a new sentence with "However"

line 235 0.1 (what?)

line 240 change "attempted" to "used"; add "period" after "reaction" and start a new sentence with "However"

line 249 should "poffer" be "prove"? Capitallise "t""

line 280 use either " production of zoospores" or "zoosporgenis"

line 283. any literature might be cited for the procedure used?

line 350. delete "the production of the motile life cycle stage of oomycetes". readers should know the term "zoosporenesis" by now

line 411. delete "though should be seen" and "the" after "in"

line 429 delete "of"

line 451. add citation no. after the authors.

line 453. than (what?) , citatitation no. alone is no enough.

line ";" should be ":". Capitalize "t"

line 611, 655. what is the approximate temperature range for "room temperature"?

line 670 delete "Caution : vigorous reaction"

Author Response

The authors would like to thank the reviewer for their comments on our manuscript and have endeavoured to address each one in specific in the revised manuscript. The response to each specific comment is outlined below in red italics:

Reviewer 2

In general, the MS was well presented. The authors did successfully carried out the experiment and came out with some significant and interesting results. The literature cited was updated. However, they were a lit inconsistent in using the past tense in methods and results. They should all be in past tense. Further, there are some minor points that can be improved:

line 71 add citation reference no. after the authors. Moved reference 15 to this position

line 84 place "plant genus" before "Cinnamona" Rephrased this as directed

line 88 add citation no. after "Peligot" Split up double reference 21,22 and moved 21 as directed

line 85 add citation no. after "Quintanilla" Moved reference up and adapted text to match

line 99 Insert "comma" after " formation"; delete "new" Inserted comma and rephrased the end of the sentence appropriately.

line 101 delete (pathogenic skin fungi), readers should know the term "dermatophytes" Deleted as requested

line 120 & 121. too much duplication. Either use "sporogenesis and zoosporogenesis" or simply " mycelial growth, sporangial formation and zoosporpore production as well as germination" Rephrased with a more simple definition as directed

line 193 delete "hyphae"; "mycelia" alone will do Deleted as requested

line 196 delete "of" Deleted as directed

line 197 add "comma" after "incubation" Added as directed

line 202 change "dinstinct" to "different" Changed distinct to different

line 234. add "period" after "material" and start a new sentence with "However" Not changed – this change would not make grammatical sense.

line 235 0.1 (what?) Not changed – equivalent is a common chemical term to label a proportion or quantity added to a reaction.

line 240 change "attempted" to "used"; add "period" after "reaction" and start a new sentence with "However" Changed attempted to used; did not change reaction or however as it does not make grammatical sense.

line 249 should "poffer" be "prove"? Capitallise "t"" Not changed - proffer means offer or give; the convention is not to capitalise trans but the structure it refers to.

line 280 use either " production of zoospores" or "zoosporgenis" Removed (zoosporogenesis)

line 283. any literature might be cited for the procedure used? References [42] and [52,53] have been added here     

line 350. delete "the production of the motile life cycle stage of oomycetes". readers should know the term "zoosporenesis" by now Deleted as directed

line 411. delete "though should be seen" and "the" after "in" Deleted as directed and rephrased sentence accordingly.

line 429 delete "of" Deleted as directed

line 451. add citation no. after the authors. Citation to reference [42] incorporated

line 453. than (what?) , citatitation no. alone is no enough. Incorporated “cinnamaldehyde” before (7)

line ";" should be ":". Capitalize "t" No line number here so could not address the comment – assuming the t refers to trans-Cinnamaldehyde, this is not changed in line with convention.

line 611, 655. what is the approximate temperature range for "room temperature"? A line to address this has been added to the general Materials and Methods

line 670 delete "Caution : vigorous reaction" Not changed – this is the convention for reporting the practice of a chemical reaction where caution must be used to prevent potential accident.

Thank you very much for your comments

Reviewer 3 Report

The manuscript describes the synthesis of new cinnamaldehyde and ellipticine derivatives and their efficacy against the Phytophthora infestans. Also, the cytotoxicity was evaluated. The manuscript is informative but I would like to address some commentaries.

Abstract L22-23 - " An ellipticine derivative which incorporates an aldehyde has been identified with exceptional activity versus P. infestans with limited toxicity and potential for use as a fungicide." Mention from the very beginning which is the compound.

The Introduction section is too long, even it is well-written. In my opinion, it could be split. A more concise introduction for the present paper and the rest of the text could be further developed for a review.

The number of compounds tested should be clearly written from the very beginning in order to easily to be followed the results and the Figures.

L384-385 is clearly written - "seven cinnamaldehyde, five ellipticines in addition to six stock aldehydes" In figure 13 - there is one aldehyde missing?

L2923-295. The sentence is confusing. At least for me.

L301- the presence of the aldehyde group is present only in the cinnamaldehyde (7) compound? More discussion about the connection of the aldehyde group and inhibitory effect against P. infestans. A more concise explanation about the connection between structural properties and antifungal and cytotoxic effects would be welcome.

L436-438 "It is postulated the ..." Are some references which could sustain the hypothesis?

L428 - the authors identify the indol as a relevant detail. More explanation about, for the readers not familiar with chemical structures.

L471 - if the concentration of  25 microM is not the best choice, why the authors still designed their study accordingly?

Author Response

The authors would like to thank the reviewer for their comments on our manuscript and have endeavoured to address each one in specific in the revised manuscript. The response to each specific comment is outlined below in red italics:

Reviewer 3

The manuscript describes the synthesis of new cinnamaldehyde and ellipticine derivatives and their efficacy against the Phytophthora infestans. Also, the cytotoxicity was evaluated. The manuscript is informative but I would like to address some commentaries.

Abstract L22-23 - " An ellipticine derivative which incorporates an aldehyde has been identified with exceptional activity versus P. infestans with limited toxicity and potential for use as a fungicide." Mention from the very beginning which is the compound. (9-formyl-6-methylellipticine, 34) has been added to the abstract to identify the new lead compound

The Introduction section is too long, even it is well-written. In my opinion, it could be split. A more concise introduction for the present paper and the rest of the text could be further developed for a review. Not changed – while the reviewer makes an excellent point, this manuscript contains synthetic chemistry, the biology of plant and mammalian cells and should cover the development of fungicidal agents against P. Infestans to date. We have submitted a follow-on paper in which the introductory section is significantly reduced and it is intended that these two papers be read in tandem and hence the longer introduction here is balanced by the second paper. If insisted on by the reviewer/editorial team we will happily shorten the introductory section but would prefer not to.

The number of compounds tested should be clearly written from the very beginning in order to easily to be followed the results and the Figures. The phrase “In total, seventeen compounds…..” has been added to L289 at the beginning of the biological evaluation to clarify this.

L384-385 is clearly written - "seven cinnamaldehyde, five ellipticines in addition to six stock aldehydes" In figure 13 - there is one aldehyde missing? The structure of compound 19 is included for reference in order to highlight the influence of indole-3-aldehyde natural products in plants and was not subsequently tested but substituted derivatives were. A statement to this effect has been included on L226. “Indole-3-aldehydes 20 and 21 are structurally related to the tryptophan-derived plant metabolite parent (19) which has been identified to confer immunity in plants to microbial pathogen infection.” The reference to six stock aldehydes changed to five on L384.

L2923-295. The sentence is confusing. At least for me. This has been rephrased as “All cinnamaldehyde derivatives 17, 18, 26-30 prove not as potent as the parent cinnamaldehyde (7), although this is tested at a much higher concentration”

L301- the presence of the aldehyde group is present only in the cinnamaldehyde (7) compound? More discussion about the connection of the aldehyde group and inhibitory effect against P. infestans. A more concise explanation about the connection between structural properties and antifungal and cytotoxic effects would be welcome. The aldehyde group is a consistent reference between the compounds we have assayed. This section has been rewritten in the R&D to explore the relevance of this further:

As expected, cinnamaldehyde (7) shows good inhibition of P. infestans which given its simple structure has been postulated to arise from the presence of an a,b-unsaturated aldehyde group and its potential to interact with cysteines in target proteins. Compounds 7, 17, 18, 20-24 and 34-36 arise from different chemical classes but all contain an aldehyde and in comparing their activity, there is no clear relationship between this functional group and mycelial growth.

The functionalised cinnamaldehydes 17 and 18 which contain the a,b-unsaturated aldehyde are identified with relatively low mycelial growth inhibition within this screen. Each of the proprietary aromatic aldehydes tested (20–24) show moderate to low inhibition of P. infestans with 5-bromoindole-3-carbaldehyde (21) seen as the most promising.

The most effective compounds in the screen are cinnamaldehyde (7) and 9-formyl-6-methylellipticine (34) and structurally these comprise an a,b-unsaturated aldehyde and an aromatic aldehyde. While it is evidently important for potency, this suggests that the aldehyde group is not solely responsible for inhibition of mycelial growth and that other factors must be explored.

Also added following phrase under cytotoxicity results (2.4.3.1):

“Given the electrophilic nature of the screening panel of compounds and potential for bioadduct formation, the establishment of relative toxicity is an important step in the development towards lead compound choice and potential for field trials.”

L436-438 "It is postulated the ..." Are some references which could sustain the hypothesis? A reference has been included to the bioactivity of ellipticine derivatives which include aldehyde and alkyl substitutions.

L428 - the authors identify the indol as a relevant detail. More explanation about, for the readers not familiar with chemical structures. Sentence added to reflect the importance of the indole “Indole derivatives and indole-3-carbaldehydes are natural products derived from the amino acid tryptophan and have been shown to provide innate plant resistance against pathogens so this deserves consideration for future investigations”

L471 - if the concentration of  25 microM is not the best choice, why the authors still designed their study accordingly? The reviewer raises an excellent point however this manuscript should be seen in the light that it is the first of a two part series where in the second manuscript (also submitted to this special edition), the ellipticines discovered in this work and the optimal dosage are focussed on in more detail. The work presented here clearly identifies the activity of new lead compounds against P. Infestans in a screen at 25uM and this initial dose was chosen as one significantly lower than both control compounds used: the natural product cinnamaldehyde (1mM) and the marketed fungicide mancozeb (100uM) and so would indicate potential for field trials.

Thank you very much for your comments

Round 2

Reviewer 3 Report

Dear authors,

I am really appreciated your experimental and writing work. You have responded to my comments.

The MDPI editor does not restrict the length of the manuscripts, but there are some general recommendations about the length of the introduction part. Even my opinion is different from yours if the editors appreciate that the proportion of the manuscript parts is acceptable I do not have further comments.

As I understand, the results presented in the present paper are part of a bigger study, and this point should be clearly written in the last paragraph of the introduction or presented as a further perspective view in the R&D part.

Good luck with your research

Author Response

We thank the reviewer again for their very constructive comments. In order to address their concerns we have added the following text to the manuscript:

L182 (end of introduction): Given the experience of the group, of particular interest are the ellipticines which would offer a new framework for the inhibition P. infestans growth and the potential for overcoming resistance.

L784(end of conclusion): However, the discovery of 9-formyl-6-methylellipticine 34 as a potent growth inhibitor is the subject of our investigation into the influence of 2- and 6-alkylated 9-formylellipticines on P. infestans growth [64].  

New reference to extended study added:

[64] M.L. McKee, L. Zheng, E.C. O’Sullivan, R.A. Kehoe, B.M. Doyle, J.J. Mackrill, F.O. McCarthy, Synthesis and evaluation of novel ellipticines and derivatives as inhibitors of Phytophthora infestans. Pathogens (2020) under review.

Thank you again for your comments.